# Nasopharyngeal Carcinoma Cell Lines: Reliable Alternatives to Primary Nasopharyngeal Cells?

**DOI:** 10.3390/cells13070559

**Published:** 2024-03-22

**Authors:** Anna Makowska, Ralf Weiskirchen

**Affiliations:** 1Division of Pediatric Hematology, Oncology and Stem Cell Transplantation, RWTH University Hospital Aachen, D-52074 Aachen, Germany; 2Institute of Molecular Pathobiochemistry, Experimental Gene Therapy and Clinical Chemistry (IFMPEGKC), RWTH University Hospital Aachen, D-52074 Aachen, Germany

**Keywords:** nasopharyngeal carcinoma, HeLa cell line, cross-contamination, STR profiling

## Abstract

Nasopharyngeal carcinoma (NPC) is a type of cancer that originates from the mucosal lining of the nasopharynx and can invade and spread. Although contemporary chemoradiotherapy effectively manages the disease locally, there are still challenges with locoregional recurrence and distant failure. Therefore, it is crucial to have a deeper understanding of the molecular basis of NPC cell movement in order to develop a more effective treatment and to improve patient survival rates. Cancer cell line models are invaluable in studying health and disease and it is not surprising that they play a critical role in NPC research. Consequently, scientists have established around 80 immortalized human NPC lines that are commonly used as in vitro models. However, over the years, it has been observed that many cell lines are misidentified or contaminated by other cells. This cross-contamination leads to the creation of false cell lines that no longer match the original donor. In this commentary, we discuss the impact of misidentified NPC cell lines on the scientific literature. We found 1159 articles from 2000 to 2023 that used NPC cell lines contaminated with HeLa cells. Alarmingly, the number of publications and citations using these contaminated cell lines continued to increase, even after information about the contamination was officially published. These articles were most commonly published in the fields of oncology, pharmacology, and experimental medicine research. These findings highlight the importance of science policy and support the need for journals to require authentication testing before publication.

## 1. Introduction

Nasopharyngeal carcinoma (NPC) is a cancer that originates from the mucosal lining of the nasopharynx and can spread to other parts of the body. This type of tumor is most commonly found in East and Southeast Asia, particularly in the southern regions of China [1]. In China, approximately 3 out of every 100,000 people are diagnosed with NPC each year, while globally the number is 1.5 out of 100,000. The male to female ratio is around two-to-five [2,3,4,5]. It was estimated that there were 80,008 deaths from NPC in 2020. By the year 2040, an increase in the annual number of NPC cases is expected, rising from 133,354 in 2020 to 179,476. The projected number of deaths is expected to increase from 80,008 to 113,851. This represents a 34.58% increase in cases and a 42.29% increase in deaths, compared to 2020 [5]. Possible causes of NPC include exposure to secondhand smoke, alcohol consumption, eating preserved foods, infection with the Epstein–Barr virus (EBV), and genetic variations in the host [6]. According to the World Health Organization (WHO), NPC is divided into keratinizing squamous cell carcinoma (WHO type I); differentiated, non-keratinizing carcinoma (WHO type II); and undifferentiated, non-keratinizing carcinoma (WHO type III) [7].

Despite advances in chemoradiotherapy, both the recurrence of the tumor in the local area and its spread to distant sites remain significant challenges [8]. It is estimated that 21.3% of patients with nonmetastatic NPC experience treatment failure, primarily due to distant metastasis, within five years of initial treatment [9]. Therefore, a better understanding of the molecular mechanisms underlying the movement of NPC cells is crucial for developing more effective treatment strategies and improving patient survival rates.

Cancer cell line models serve as an essential laboratory resource, simplifying the study of health and disease. The majority of cell-intrinsic, targeted therapeutics were discovered and developed using continuous cell lines. These cell lines can be passaged repeatedly, recover reliably from cryopreservation, and maintain many characteristics of their original cell type or tissue [10,11,12]. These benefits make continuous cell lines an efficient and popular model system for studying normal cellular processes and various disease conditions.

For our study, we conducted two separate searches on the PubMed database. Both searches took place on 7 March 2024. The first search, using the terms “nasopharyngeal carcinoma” and “cell line”, resulted in 3746 hits for the period from 1970 to 2024 (https://pubmed.ncbi.nlm.nih.gov/?term=%28%28nasopharingeal+carcinoma%29+AND+%28cell+line%29%29; accessed on 7 March 2024). The second search used the terms “nasopharyngeal carcinoma” and “cell line” for each single year between 1979 and 2024 (Table 1).

This second search yielded 4365 hits for the same period as the first search. This indicates an increasing use of NPC cell line models over the years, as evidenced by the number of studies. For instance, there were 17 studies in 1985, 78 studies in 2005, and 270 studies in 2020 (Figure 1).

Furthermore, a search for “nasopharyngeal carcinoma cell” conducted on 15 February 2024 on the Cellosaurus database, which is widely recognized as an authoritative source for cell line information, resulted in 80 human NPC cell lines (Table 2).

Cell lines play a crucial role in research, but they are often misidentified or contaminated by other cells. This can result in the creation of incorrect cell lines that do not match the original donor and may come from a different donor or species. Contamination can occur early on, where the original cell line may have never existed independently, or later, where the original may still exist in other stocks but is contaminated in the tested sample. This issue impacts the authenticity of cell lines as cancer models. Even widely used cell lines may not accurately represent the tumor type they are intended to come from due to changes in cell line shape, growth, and gene expression. Neglecting this problem can lead to inaccurate and misleading research data [57].

In cancer research, cell lines serve as invaluable models for studying cancer progression, diagnostics, and treatment advancements. Although they may not perfectly mimic the original tumor, they generally exhibit similar characteristics. Low-passage cancer cell lines are preferred due to their closer resemblance to the original tumor. Prolonged culturing can alter genetic and molecular profiles, emphasizing the need to monitor passage numbers. Variations in proliferation, migration, gene expression, and drug sensitivity can be attributed to passage number differences. Additionally, passage number influences DNA methylation levels, metabolic profiles, and gene/protein expression in cancer cell lines. Researchers must carefully select the appropriate passage number to maintain the consistency and reliability of the results. To ensure dependability and replicability, scientists should actively track growth, migration, and gene/protein profiles at specific passage numbers [58]. Prolonged culture can lead to reduced or altered key functions, rendering cell lines unreliable models of their original source material [59].

For nearly half a century, sporadic reports of HeLa cell contamination in numerous mammalian cell lines have been documented [60,61,62,63]. The previously mentioned contaminated cell lines have short tandem repeat (STR) profiles that are either identical or highly similar to HeLa. STR profiling is the most popular method for detecting cell contamination (Table 3). This analysis determines a cell line’s DNA profile by counting the number of times a DNA sequence, known as an STR unit, appears at a specific chromosomal location. The chromosomal location refers to the specific area on a chromosome where the DNA strand is located [64]. This molecular genotyping technique is now available for both human and mouse cell lines. When used regularly in cell culture management, this technique, along with other methods, significantly improves the detection of cellular cross-contamination. This leads to more reproducible and scientifically meaningful research outcomes [65].

The American Tissue Culture Collection (ATCC) has established standard guidelines advocating for the use of at least eight STR loci in human cell line authentication. These recommended loci include TH01, TPOX, vWA, CSF1PO, D16S539, D7S820, D13S317, and D5S8181, along with the Amelogenin gene for gender identification [57,66]. These guidelines set the standards for authenticating cell lines. These STR loci play a crucial role in ensuring the quality and integrity of human cell lines, allowing researchers to demonstrate relatedness between cell lines and uniquely identify human cells. The adoption of these guidelines has facilitated broader cell line authentication testing in laboratories worldwide.

**Table 3 cells-13-00559-t003:** Cell contamination detection methods.

Detection Method	Description	Reference
Isoenzyme analysis	The approach relies on the isoelectric separation of a distinct group of intracellular enzymes. These enzymes serve as markers, allowing differentiation between cell lines originating from humans, mice, or other mammals.	[67]
HLA typing	HLA typing, which uses serological methods and specific sera, identifies HLA antigens on cell surfaces. This complementary test helps detect cross-contamination within cell lines of the same species.	[68]
DNA fingerprinting	DNA fingerprinting relies on determining specific DNA sequences. Variable Number of Tandem Repeat (VNTR) and Short Tandem Repeat (STR) loci are amplified using PCR.	[69]

The first instance of cross-contamination in NPC cell lines was reported in 2008 by Chan and his team, who discovered a significant degree of similarity between the STR profiles of two EBV-negative cell lines, CNE-1 and CNE-2 [31]. Additionally, they found that the STR profile of HeLa perfectly matched the STR profiles of CNE-1 and CNE-2 listed in the ATCC [31]. In 2015, Ye and his colleagues published a list of additional NPC cell lines contaminated with HeLa cells [54]. The recently released 12th version of the International Cell Line Authentication Committee (ICLAC) lists a total of eight NPC cell lines as being cross-contaminated (Table 4) [57].

All cell lines in this group carry the 13.3 allele at the STR locus D13S317, indicating that they are derived from HeLa, but also contain additional genetic material from an unidentified source. The true identities of the cell lines in this group remain uncertain. In addition, many other NPC cell lines listed in the Cellosaurus database (but not in the ICLAC register) are derived from the parental cells mentioned earlier, such as 13-9B (parental cell line SUNE), 5-8F/Erbitux (parental cell line 5-8F), CNE-2Z (parental cell line CNE-2), CNE1 SRPK1 KO, CNE1 SRPK1/2 KO, and CNE1 SRPK2 KO (all derived from parental cell line CNE-1), as well as CNE-IR (parental cell line CNE-2), HNE-2 (almost identical to HNE-1), HONE-1/CPT30 and HONE-1/CPT30R (parental cell line HONE-1), S18, S18-1C3, and S26 (parent cell line CNE-2). Considering all NPC cell lines mentioned earlier, it is reasonable to speculate that at least 25% of all NPC cells should be classified as misidentified.

Despite genetic profiling evidence showing that these cell lines are contaminated with HeLa cells, numerous research groups studying NPC have continued to use these cells, up until the present time. We conducted searches on the PubMed database from 2000 to 2023 to identify papers that used the HeLa-contaminated NPC cell lines listed in Table 3. Our search specifically targeted the improper use of these cell lines, which we defined as any description that was inaccurate or misleading when using the terms “nasopharyngeal carcinoma cell line”. We determined the incorrect usage of these cells based on the description provided, as these descriptions guide researchers in selecting cell lines for their studies. The search involved examining titles, abstracts, and the text of articles. Our study primarily focused on titles, abstracts, and methods sections, often without reference to the main text. Using this method, we identified a total of 1282 articles from 2000 to 2023 that utilized NPC cell lines contaminated with HeLa cells (Figure 2).

While the number of journals requiring authentication is increasing, only a few journal articles currently provide information on authentication testing. Researchers have no choice but to rely on the cell line data provided by the authors in order to determine the suitability of a cell line for their research. Although the tissue type or disease condition may not appear relevant to all studies, readers will use this information to select their own cell lines. Therefore, the information about cell lines must be accurate; otherwise, it can lead to misunderstandings within the research community when interpreting the results of that work. This issue becomes particularly crucial when cell line models are utilized for testing therapeutic agents [72].

A search conducted on the “Web of Science” database on 22 January 2024, using the terms “nasopharyngeal carcinoma” and “one of the eight HeLa contaminated cell lines”, reveals a growing trend in the use of misidentified NPC cell lines as models for NPC, over the years. Furthermore, the number of citations for these erroneous publications is also on the rise (as shown in Figure 3 and Table 5).

A concerning fact is that the number of publications and citations of misidentified NPC cell lines continued to increase even after the official announcement that NPC cell lines were contaminated with HeLa cell lines [31,54]. To better understand this issue, we examined information for the CNE2 cell line. This line is one of the most commonly used NPC cell lines identified as HeLa-contaminated. CNE2 is a poorly differentiated NPC epithelial cell line that originated from a primary tumor biopsy in China in 1983 [24]. Since most NPCs are poorly differentiated, the CNE2 cell line has been a crucial model for nasopharyngeal studies. A search on the “Web of Science” database using the terms “CNE2” and “nasopharyngeal carcinoma” yielded 416 articles focusing on the CNE2 cell line from 1983 to 2023. This cell line was identified as HeLa-contaminated in 2008 [31]. From 2008 to 2023, this cell line was mentioned in 361 articles. Furthermore, these articles were cited in an additional 5271 articles, including 132 articles with self-citations. Almost all articles did not provide an authentication test. For many decades, CNE2 has been a fundamental tool in various fields of biomedical research. Numerous grants have been allocated and significant progress has been made in understanding disease etiology and developing treatments and cures.

A search on the “Web of Science” database using the terms “CNE2” and “nasopharyngeal carcinoma” revealed the most prevalent research fields, including oncology, pharmacology, and experimental medical research. These fields were identified based on articles that focused on the CNE2 cell line from 2008 (the year it was discovered that CNE2 is contaminated with HeLa) to 2023 (as shown in Figure 4).

This search also brought attention to the global issue of using cells contaminated with HeLa. Assuming that approximately 10–12% of all NPC cell lines used worldwide are cross-contaminated, this suggests that potentially hundreds of millions of dollars spent in the biomedical and healthcare sectors globally may have been wasted on producing inaccurate or misleading data.

The examination of the eight NPC cell lines contaminated with HeLa presents a multifaceted scenario with various descriptions. Often, authors fail to acknowledge that NPC cells are actually variants of HeLa, even when they cite accurate sources that have reported this information. Furthermore, they tend to provide details about the source laboratory without including literature references that would help clarify the characteristics of the cells.

So far, improvements in citations have been lagging behind publishing changes, leading to a significant amount of incorrect data persisting in the research literature. Recent initiatives, such as standardizing cell line authentication [57], promoting consistent terminology [74], encouraging authentication in journals [75], establishing the International Cell Line Authentication Committee [76], and creating Cellosaurus, a cell line database [77], could help to increase the number of papers accurately citing NPCs.

Accurate cell line data and testing are crucial. However, even with authentication, cell lines may not accurately represent their original tissues or diseases due to factors such as source and reagent differences [78,79]. Therefore, the validity of a cell line as a research model, not only in the field of NPCs, should always be continually improved [80,81].

## 2. Conclusions

Improving the selection, cultivation conditions, and reporting practices of cell lines will enhance our preclinical research models. However, a significant concern arises if the volume of publications containing incorrect information about certain cell lines exceeds those with accurate data. This situation leads to the continued use of these erroneous cell lines in scientific research, despite their inaccuracies. They are still being referenced in the literature, which is a testament to the gravity of this issue. This problem is not just a fleeting one, but rather a persistent issue that could continue to plague the scientific community for the foreseeable future. The continued use of these incorrect cell lines could potentially skew research findings and conclusions. This is not a trivial matter as it could lead to a ripple effect of misinformation within the field. Misinformation can spread like wildfire, leading to a cascade of flawed research that builds upon the incorrect data. This could potentially result in wasted resources, misguided research directions, and, worst of all, incorrect scientific conclusions that could have far-reaching implications. Therefore, it is crucial to address this issue to ensure the integrity and reliability of future research. It is not just about correcting the current state of affairs, but also about setting up robust systems and practices to prevent such issues from recurring. This includes stringent verification processes, rigorous peer reviews, and fostering a culture of transparency and accountability in the scientific community. The routine use of STR profiling, isoenzyme analysis, HLA typing, DNA fingerprinting, or any other methods that allow for the detection of cross-contamination or misidentification of a cell line should be mandatory in all work with cell lines. By implementing these measures, we can safeguard the future of scientific research, ensuring that it remains a reliable source of knowledge and insight for humanity.

## Figures and Tables

**Figure 1 cells-13-00559-f001:**
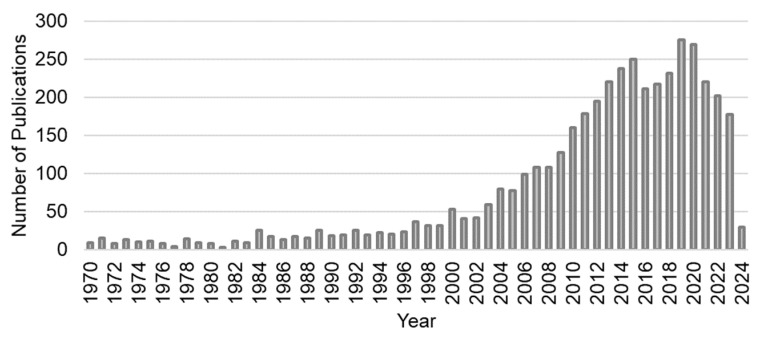
Number of nasopharyngeal carcinoma (NPC) studies. Usage of NPC cell lines in studies published between 1970 and 2024. The data are based on a PubMed search conducted on 7 March 2024.

**Figure 2 cells-13-00559-f002:**
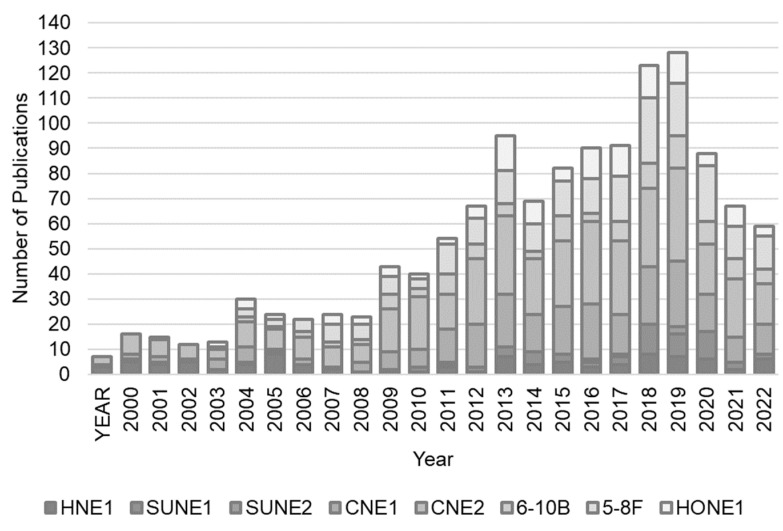
Nasopharyngeal carcinoma cells (NPCs) contaminated with HeLa cell line. Usage of NPC cell lines contaminated with HeLa cells were found in studies published from 2000 to 2023. The data are derived from a PubMed search conducted on 22 January 2024.

**Figure 3 cells-13-00559-f003:**
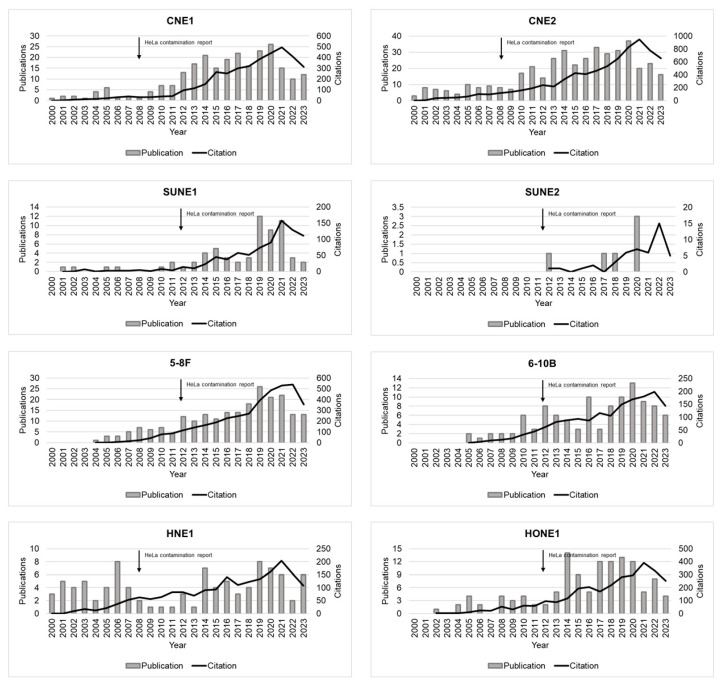
Number cited and publications over time. This figure illustrates the distribution of articles that incorrectly described nasopharyngeal carcinoma cells (represented by grey bars) and the distribution of citations of these articles (represented by the black line) from 2000 to 2023. The data were obtained from Web of Science [73].

**Figure 4 cells-13-00559-f004:**
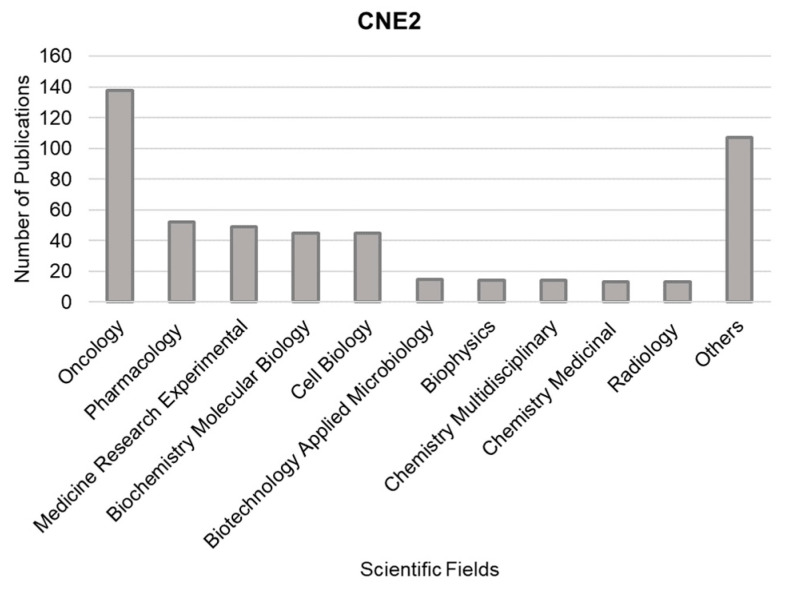
This analysis covers scientific fields featured in articles that utilize the CNE2 cell line, as well as the distribution of these fields in articles published between 2000 and 2023. The data were derived from the Web of Science [73]. Only papers with the term “CNE2” and not “CNE-2” were considered in this analysis.

**Table 1 cells-13-00559-t001:** Number of in vitro nasopharyngeal carcinoma (NPC) studies categorized by year.

Year	Uniform Resource Locator (URL) for Analyzed Year
1970	https://pubmed.ncbi.nlm.nih.gov/?term=%28%28nasopharingeal+carcinoma%29+AND+%28cell+line%29%29+AND+%281970%29
1971	https://pubmed.ncbi.nlm.nih.gov/?term=%28%28nasopharingeal+carcinoma%29+AND+%28cell+line%29%29+AND+%281971%29
1972	https://pubmed.ncbi.nlm.nih.gov/?term=%28%28nasopharingeal+carcinoma%29+AND+%28cell+line%29%29+AND+%281972%29
1973	https://pubmed.ncbi.nlm.nih.gov/?term=%28%28nasopharingeal+carcinoma%29+AND+%28cell+line%29%29+AND+%281973%29
1974	https://pubmed.ncbi.nlm.nih.gov/?term=%28%28nasopharingeal+carcinoma%29+AND+%28cell+line%29%29+AND+%281974%29
1975	https://pubmed.ncbi.nlm.nih.gov/?term=%28%28nasopharingeal+carcinoma%29+AND+%28cell+line%29%29+AND+%281975%29
1976	https://pubmed.ncbi.nlm.nih.gov/?term=%28%28nasopharingeal+carcinoma%29+AND+%28cell+line%29%29+AND+%281976%29
1977	https://pubmed.ncbi.nlm.nih.gov/?term=%28%28nasopharingeal+carcinoma%29+AND+%28cell+line%29%29+AND+%281977%29
1978	https://pubmed.ncbi.nlm.nih.gov/?term=%28%28nasopharingeal+carcinoma%29+AND+%28cell+line%29%29+AND+%281978%29
1979	https://pubmed.ncbi.nlm.nih.gov/?term=%28%28nasopharingeal+carcinoma%29+AND+%28cell+line%29%29+AND+%281979%29
1980	https://pubmed.ncbi.nlm.nih.gov/?term=%28%28nasopharingeal+carcinoma%29+AND+%28cell+line%29%29+AND+%281980%29
1981	https://pubmed.ncbi.nlm.nih.gov/?term=%28%28nasopharingeal+carcinoma%29+AND+%28cell+line%29%29+AND+%281981%29
1982	https://pubmed.ncbi.nlm.nih.gov/?term=%28%28nasopharingeal+carcinoma%29+AND+%28cell+line%29%29+AND+%281982%29
1983	https://pubmed.ncbi.nlm.nih.gov/?term=%28%28nasopharingeal+carcinoma%29+AND+%28cell+line%29%29+AND+%281983%29
1984	https://pubmed.ncbi.nlm.nih.gov/?term=%28%28nasopharingeal+carcinoma%29+AND+%28cell+line%29%29+AND+%281984%29
1985	https://pubmed.ncbi.nlm.nih.gov/?term=%28%28nasopharingeal+carcinoma%29+AND+%28cell+line%29%29+AND+%281985%29
1986	https://pubmed.ncbi.nlm.nih.gov/?term=%28%28nasopharingeal+carcinoma%29+AND+%28cell+line%29%29+AND+%281986%29
1987	https://pubmed.ncbi.nlm.nih.gov/?term=%28%28nasopharingeal+carcinoma%29+AND+%28cell+line%29%29+AND+%281987%29
1988	https://pubmed.ncbi.nlm.nih.gov/?term=%28%28nasopharingeal+carcinoma%29+AND+%28cell+line%29%29+AND+%281988%29
1989	https://pubmed.ncbi.nlm.nih.gov/?term=%28%28nasopharingeal+carcinoma%29+AND+%28cell+line%29%29+AND+%281989%29
1990	https://pubmed.ncbi.nlm.nih.gov/?term=%28%28nasopharingeal+carcinoma%29+AND+%28cell+line%29%29+AND+%281990%29
1991	https://pubmed.ncbi.nlm.nih.gov/?term=%28%28nasopharingeal+carcinoma%29+AND+%28cell+line%29%29+AND+%281991%29
1992	https://pubmed.ncbi.nlm.nih.gov/?term=%28%28nasopharingeal+carcinoma%29+AND+%28cell+line%29%29+AND+%281992%29
1993	https://pubmed.ncbi.nlm.nih.gov/?term=%28%28nasopharingeal+carcinoma%29+AND+%28cell+line%29%29+AND+%281993%29
1994	https://pubmed.ncbi.nlm.nih.gov/?term=%28%28nasopharingeal+carcinoma%29+AND+%28cell+line%29%29+AND+%281994%29
1995	https://pubmed.ncbi.nlm.nih.gov/?term=%28%28nasopharingeal+carcinoma%29+AND+%28cell+line%29%29+AND+%281995%29
1996	https://pubmed.ncbi.nlm.nih.gov/?term=%28%28nasopharingeal+carcinoma%29+AND+%28cell+line%29%29+AND+%281996%29
1997	https://pubmed.ncbi.nlm.nih.gov/?term=%28%28nasopharingeal+carcinoma%29+AND+%28cell+line%29%29+AND+%281997%29
1998	https://pubmed.ncbi.nlm.nih.gov/?term=%28%28nasopharingeal+carcinoma%29+AND+%28cell+line%29%29+AND+%281998%29
1999	https://pubmed.ncbi.nlm.nih.gov/?term=%28%28nasopharingeal+carcinoma%29+AND+%28cell+line%29%29+AND+%281999%29
2000	https://pubmed.ncbi.nlm.nih.gov/?term=%28%28nasopharingeal+carcinoma%29+AND+%28cell+line%29%29+AND+%282000%29
2001	https://pubmed.ncbi.nlm.nih.gov/?term=%28%28nasopharingeal+carcinoma%29+AND+%28cell+line%29%29+AND+%282001%29
2002	https://pubmed.ncbi.nlm.nih.gov/?term=%28%28nasopharingeal+carcinoma%29+AND+%28cell+line%29%29+AND+%282002%29
2003	https://pubmed.ncbi.nlm.nih.gov/?term=%28%28nasopharingeal+carcinoma%29+AND+%28cell+line%29%29+AND+%282003%29
2004	https://pubmed.ncbi.nlm.nih.gov/?term=%28%28nasopharingeal+carcinoma%29+AND+%28cell+line%29%29+AND+%282004%29
2005	https://pubmed.ncbi.nlm.nih.gov/?term=%28%28nasopharingeal+carcinoma%29+AND+%28cell+line%29%29+AND+%282005%29
2006	https://pubmed.ncbi.nlm.nih.gov/?term=%28%28nasopharingeal+carcinoma%29+AND+%28cell+line%29%29+AND+%282006%29
2007	https://pubmed.ncbi.nlm.nih.gov/?term=%28%28nasopharingeal+carcinoma%29+AND+%28cell+line%29%29+AND+%282007%29
2008	https://pubmed.ncbi.nlm.nih.gov/?term=%28%28nasopharingeal+carcinoma%29+AND+%28cell+line%29%29+AND+%282008%29
2009	https://pubmed.ncbi.nlm.nih.gov/?term=%28%28nasopharingeal+carcinoma%29+AND+%28cell+line%29%29+AND+%282009%29
2010	https://pubmed.ncbi.nlm.nih.gov/?term=%28%28nasopharingeal+carcinoma%29+AND+%28cell+line%29%29+AND+%282010%29
2011	https://pubmed.ncbi.nlm.nih.gov/?term=%28%28nasopharingeal+carcinoma%29+AND+%28cell+line%29%29+AND+%282011%29
2012	https://pubmed.ncbi.nlm.nih.gov/?term=%28%28nasopharingeal+carcinoma%29+AND+%28cell+line%29%29+AND+%282012%29
2013	https://pubmed.ncbi.nlm.nih.gov/?term=%28%28nasopharingeal+carcinoma%29+AND+%28cell+line%29%29+AND+%282013%29
2014	https://pubmed.ncbi.nlm.nih.gov/?term=%28%28nasopharingeal+carcinoma%29+AND+%28cell+line%29%29+AND+%282014%29
2015	https://pubmed.ncbi.nlm.nih.gov/?term=%28%28nasopharingeal+carcinoma%29+AND+%28cell+line%29%29+AND+%282015%29
2016	https://pubmed.ncbi.nlm.nih.gov/?term=%28%28nasopharingeal+carcinoma%29+AND+%28cell+line%29%29+AND+%282016%29
2017	https://pubmed.ncbi.nlm.nih.gov/?term=%28%28nasopharingeal+carcinoma%29+AND+%28cell+line%29%29+AND+%282017%29
2018	https://pubmed.ncbi.nlm.nih.gov/?term=%28%28nasopharingeal+carcinoma%29+AND+%28cell+line%29%29+AND+%282018%29
2019	https://pubmed.ncbi.nlm.nih.gov/?term=%28%28nasopharingeal+carcinoma%29+AND+%28cell+line%29%29+AND+%282019%29
2020	https://pubmed.ncbi.nlm.nih.gov/?term=%28%28nasopharingeal+carcinoma%29+AND+%28cell+line%29%29+AND+%282020%29
2021	https://pubmed.ncbi.nlm.nih.gov/?term=%28%28nasopharingeal+carcinoma%29+AND+%28cell+line%29%29+AND+%282021%29
2022	https://pubmed.ncbi.nlm.nih.gov/?term=%28%28nasopharingeal+carcinoma%29+AND+%28cell+line%29%29+AND+%282022%29
2023	https://pubmed.ncbi.nlm.nih.gov/?term=%28%28nasopharingeal+carcinoma%29+AND+%28cell+line%29%29+AND+%282023%29
2024	https://pubmed.ncbi.nlm.nih.gov/?term=%28%28nasopharingeal+carcinoma%29+AND+%28cell+line%29%29+AND+%282024%29

All links were accessed on 7 March 2024.

**Table 2 cells-13-00559-t002:** Immortalized human nasopharyngeal carcinoma cell lines listed in the Cellosaurus database [13].

No.	Cell Line	Cellosaurus ID	EBV Status	Reference
1	13-9B	CVCL_C527	IN, LN	[14]
2	2-27-Ad	CVCL_IY39	IP, LN	[15]
3	5-8F	CVCL_C528	IN, LN	[14]
4	5-8F/Erbitux	CVCL_S665	IN, LN	[16]
5	6-10B	CVCL_C529	IN, LN	[14]
6	A2L	CVCL_X201	IP	[17]
7	A2L/AH	CVCL_X202	IP	[18]
8	Ad-AH	CVCL_X200	IU, LN	[18]
9	C17	CVCL_VT47	IP	[19]
10	C666	CVCL_M597	IP, LU	[20]
11	C666-1	CVCL_7949	IP, LP	[21]
12	CG1	CVCL_J445	IP	[22]
13	CNE-1	CVCL_6888	IN, LN	[23]
14	CNE-2	CVCL_6889	IP, LN	[24]
15	CNE-2Z	CVCL_6890	IN	[25]
16	CNE-3	CVCL_M598	IN	[26]
17	CNE1 SRPK1 KO	CVCL_YT53	IN	[27]
18	CNE1 SRPK1/2 KO	CVCL_YT55	IN	[27]
19	CNE1 SRPK2 KO	CVCL_YT54	IN	[27]
20	CNE2-IR	CVCL_C9B8	IN	[28]
21	Esther	CVCL_X932	UNK	[29]
22	HNE-1	CVCL_0308	IP, LN	[30]
23	HNE-2	CVCL_FA07	LN	[31]
24	HNE-3	CVCL_FA08	LN	[30]
25	HONE-1	CVCL_8706	IP, LN	[30]
26	HONE-1/CPT30	CVCL_M595	IN	[32]
27	HONE-1/CPT30R	CVCL_M596	IN	[32]
28	Ly1	CVCL_ZU83	UNK	[33]
29	Ly11	CVCL_ZU81	UNK	[33]
30	Ly2	CVCL_ZU84	UNK	[33]
31	Ly28	CVCL_ZU82	UNK	[33]
32	Maku	CVCL_1Q44	LP	[34]
33	NA-NP15	CVCL_DG71	IN, LP	NN ^1^
34	NA-NR1	CVCL_DG72	IN, LP	NN ^1^
35	NA-NR15	CVCL_DG73	IN, LP	NN ^2^
36	NP361-CDK4R24C-hTert	CVCL_B3Q4	IU, LP	[35]
37	NP361-cyclinD1-hTert	CVCL_B3Q5	IU, LP	[35]
38	NP361hTert	CVCL_B3Q3	IU, LN	[35]
39	NP39E6/E7	CVCL_F754	IP, LP	[36]
40	NP446-CDK4R24C-hTert	CVCL_B3Q6	IU, LP	[35]
41	NP446-cyclinD1-hTert	CVCL_B3Q7	IU, LP	[35]
42	NP460hTert	CVCL_X205	IU, LP	[37]
43	NP550-CDK4R24C-hTert	CVCL_B3Q8	IU, LP	[35]
44	NP550-cyclinD1-hTert	CVCL_B3Q9	IU, LP	[35]
45	NP550hTert	CVCL_B3QA	IU, LN	[35]
46	NP69SV40T	CVCL_F755	IP, LP	[36]
47	NPC-204	CVCL_A5WV	IU, LN	[38]
48	NPC268	NA ^3^	IP, LP	[39]
49	NPC-B13	CVCL_D3FG	IU, LP	[40]
50	NPC-BM00	CVCL_B3QC	IU, LN	[41]
51	NPC-BM1	CVCL_6007	UNK	[42]
52	NPC-BM29	CVCL_B3QB	IU, LN	[43]
54	NPC-KT	CVCL_X204	IU, LP	[18]
55	NPC-TW01	CVCL_6008	IP, LN	[43]
56	NPC-TW02	CVCL_6009	IU, LN	[43]
57	NPC-TW03	CVCL_6010	IP, LN	[44]
58	NPC-TW03 EBV(+)	CVCL_ZF72	IN, LP	[45]
59	NPC-TW04	CVCL_6011	IN, LN	[44]
60	NPC-TW05	CVCL_6012	IP, LN	[44]
61	NPC-TW06	CVCL_6013	IP, LN	[44]
62	NPC-TW07	CVCL_6014	IP, LN	[44]
63	NPC-TW08	CVCL_6015	IN, LN	[44]
64	NPC-TW09	CVCL_6016	IN, LN	[44]
65	NPC-TY861	CVCL_A5WW	IP, LP	[46]
66	NPC/HK1	CVCL_7084	IN, LN	[47]
67	NPC38	CVCL_UH63	IN, LN	[48]
68	NPC43	CVCL_UH64	IP, LP	[48]
69	NPC53	CVCL_UH65	IN, LN	[48]
70	Patrick	CVCL_IU34	IP, LP	[49]
71	Ramos	CVCL_0597	IN, LN	[50]
72	Ramos-AW	CVCL_2702	IN, LN	[50]
73	Ramos/B95-8	CVCL_ZU79	IN, LN	[51]
74	Ramos/NPC	CVCL_ZU80	IN, LP	[51]
75	S18	CVCL_B0U9	IN, LN	[52]
76	S18-1C3	CVCL_B0UA	IN, LN	[53]
77	S26	CVCL_B0UB	IN, LN	[52]
78	Silfere	CVCL_W936	UNK	[29]
79	SUNE1	CVCL_6946	LN	[54]
80	SUNE2	CVCL_6956	IN, LN	[55]
81	SVK-CR2	CVCL_YD67	IU, LN	[56]
82	SVK-Neo	CVCL_YD68	IU, LN	[56]

^1^ Parent cell line: NPC-TW01. ^2^ Parent cell line: NA-NR1. ^3^ This cell line was recently established and is not yet listed in the Cellosaurus database. Abbreviations: IN, initially EBV-negative; IP, initially EBV-positive; LN, long-term culture EBV-negative; LP, long-term culture EBV-positive; LU, long-term culture EBV-status unknown; UNK, EBV status unknown.

**Table 4 cells-13-00559-t004:** Table of misidentified NPC cell lines. The list is prepared according to the International Cell Line Authentication Committee (ICLAC).

Misidentified Cell Line	ICLAC-ID	Histology	Contaminating Cell Line	Actual Species
5-8F (SUNE1 derivate)	ICLAC-00596	poorly differentiated NPC; the highest tumorigenic and metastatic ability [70]	HeLa	Human
6-10B (SUNE1 derivate)	ICLAC-00597	poorly differentiated NPC; the lowest tumorigenicity and lack of metastatic ability [13]	HeLa	Human
CNE-1 ^1^	ICLAC-00473	well-differentiated squamous cell carcinoma [22]	HeLa	Human
CNE-2 ^1^	ICLAC-00474	poorly differentiated NPC [24]	HeLa	Human
HNE-1 ^2^	not listed	poorly differentiated squamous cell carcinoma [30]	HeLa	Human
HONE-1	ICLAC-00496	poorly differentiated squamous cell carcinoma [30]	HeLa	Human
SUNE1	ICLAC-00595	poorly differentiated NPC [71]	HeLa	Human
SUNE2	ICLAC-00598	undifferentiated NPC [55]	HeLa	Human

^1^ Also known as CNE1 and CNE2. ^2^ According to the Cellosaurus database, the cell line HNE-1 (CVCL-0308) has a highly similar STR profile to the cell line HONE-1 (CVCL_8706). As both cell lines were established in the same laboratory, they must be classified as misidentified.

**Table 5 cells-13-00559-t005:** Citation report for misidentified nasopharyngeal carcinoma cells ^1^.

NPC Cell Line	Σ Publications 2000–2023	Year of HeLa Contamination Report	Σ Citing Article 2000–2023	Σ Citing Article (without Self-Citation) 2000–2023	Σ Times Cited 2000–2023 Report	Σ Times Cited (without Self-Citation) 2000–2023
5-8F	153	2015	1746	1717	1895	1850
6-10B	70	2015	723	708	769	748
CNE-1 ^2^	252	2008	2995	2926	3307	3209
CNE-2 ^2^	361	2008	5271	5139	5958	5745
HNE-1	63	2008	1020	1003	1066	1042
HONE1	123	2015	2675	2644	2937	2888
SUNE1	50	2015	472	468	491	486
SUNE2	5	2015	34	34	35	35

^1^ Data were taken from the Web of Science [73]. ^2^ Also known as CNE1 and CNE2. Summaries of the individual reports can be found at: 5-8F: https://www.webofscience.com/wos/woscc/citation-report/078cd1af-ae5b-403d-83c6-5a11501bb8d0-cad68a52 (accessed on: 22 January 2024); 6-10B: https://www.webofscience.com/wos/woscc/citation-report/8d9af056-74f5-4623-870d-a8e12e41edac-cad68016 (accessed on: 22 January 2024); CNE1: https://www.webofscience.com/wos/woscc/citation-report/3efa39d6-4463-4d0c-a51a-0a90cbd948be-c85b0591 (accessed on: 22 January 2024); CNE2: https://www.webofscience.com/wos/woscc/citation-report/a2d59c90-d979-4a24-9d54-965450e81f48-cad6478a (accessed on: 22 January 2024); HNE1: https://www.webofscience.com/wos/woscc/citation-report/cb23d688-7f35-412e-918b-cb104d4e3d0f-cad65f4f (accessed on: 22 January 2024); HONE1: https://www.webofscience.com/wos/woscc/citation-report/26af9041-724c-4ff6-adf2-4fcaa1325f98-cc481ae5 (accessed on: 22 January 2024); SUNE1: https://www.webofscience.com/wos/woscc/citation-report/fbf0dc10-a617-46bc-92b3-53420f5af93d-cad66c02 (accessed on: 22 January 2024); SUNE2: https://www.webofscience.com/wos/woscc/citation-report/9028d111-39db-445c-8950-b18b8833818e-cad67612 (accessed on: 22 January 2024).

## Data Availability

This review only presents data that have been previously published. No new data were generated.

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
