# Peer review of "Nasopharyngeal Carcinoma Cell Lines: Reliable Alternatives to Primary Nasopharyngeal Cells?"

_cells, 2024, doi:10.3390/cells13070559_

Round 1
Reviewer 1 Report
Comments and Suggestions for Authors
This commentary addresses an important issue in cancer research, namely the frequent use of misidentified or cross-contaminated cancer cell lines, with specific focus on nasopharyngeal carcinoma (NPC) cell lines. The authors have carefully analyzed the available literature and provide convincing evidence that a substantial proportion of putative NPC cell lines are either misidentified as or cross-contaminated with HeLa cells.
Although previous publications by others (cited by the authors) have reported on misidentification of cancer cells in general, or of certain NPC cell lines, the present article provides a concise analysis of the majority of available NPC cell lines, i.e. those listed in the Cellosaurus database. Thus, this article contains valuable information for NPC researchers, but is also of general interest to researchers active in other areas of cancer research.
However, the following points of criticism need to be addressed by the authors:
1) page 2, line 58: It appears that the PubMed database search for articles on nasopharyngeal carcinoma cell lines was not performed correctly. Apparently, the enormous number of 1,355,835 hits retrieved relates to the query "cell line" only, rather than to the combination of "nasopharyngeal carcinoma" and "cell line" (approx. 3,745 hits), or the combination of "nasopharyngeal carcinoma" and "in vitro study" (approx. 1,620 hits). This may have resulted from an incorrect use of "or" in the database search, and must be corrected.
This applies also to the numbers presented in Figure 1. Thus, Figure 1 must be corrected.
2) page 5, line 101: It is unclear to me how the authors reach the conclusion that "25% of all NPC cells should be classified as misidentified." Evidence is presented for approximately 10% of NPC cell lines. This issue should be clarified.
3) A short mention of STR analysis as preferred method for cell line authentication might be helpful for the reader.
Minor points:
3) page 1, line 37: The sentence "The number of deaths us also projected to increase..." must be corrected.
4) page 1, line 39: "Possibly causes..." should read as "Possible causes..."
5) page 2, lines 50+51: Also, the sentence "...can be repeatedly passaged repeatedly..." needs to be corrected.
Author Response
Dear Reviewer 1,
Thank you for taking the time to review our manuscript. Please find a pdf-file attached in which we have summarized our response to your comments and suggestions.
Regards
Ralf Weiskirchen

Reviewer 2 Report
Comments and Suggestions for Authors
The study topic is interesting and highly relevant.
The care and attention in cell cultivation and the method of work should be at the core of any in vitro experiment. Different cells should not be cultured simultaneously.
In the introduction, cite the incidence and mortality rates for women and men with NPC (Nasopharyngeal Carcinoma).
The authors should present the different cell lines for study NP since they cited different kinds of cells, such as types for carcinoma, epithelial, or EBV+. This is not clear to the reader.
Check if all cells cited in Table 1 are carcinoma cell lines “Table 1- Immortalized human nasopharyngeal carcinoma cell lines listed in the Cellosaurus.”
Describe clearly the criteria to classify NPC cell lines, according to authentication.
The cell contamination detection method is interesting to include in the table. There are several papers analyzing this topic.
The passage of the cells used is also a critical point for cellular biology. It must be mentioned because depending on how many times a cell line has been subcultured, this can perpetuate mutations and genomic alterations. About this point, the authors could present and discuss the emergence of mutations and potential changes in drug responses.
Author Response
Dear Reviewer 2,
Thank you for taking the time to review our manuscript. Please find a pdf-file attached in which we have summarized our response to your comments and suggestions.
Regards
Ralf Weiskirchen

Round 2
Reviewer 1 Report
Comments and Suggestions for Authors
The authors have satisfactorily addressed the reviewer criticism, however, the following point needs to be clarified:
On page 2, line 59, it is stated that both PubMed database searches were performed on January 22, whereas in line 62, it is stated that the first search was conducted on March 7. Please clarify and correct.
Author Response
Dear Reviewer 1,
We would like to once again express our gratitude for taking the time to review our manuscript. Your thorough review is greatly appreciated. We are thankful that you pointed out the discrepancy regarding the date. We have made the necessary correction and updated it to March 7 on line 59 of page 2.
We hope that you will agree that our work is not suitable for publication in Cells.